# Incidence of lab-confirmed dengue fever in a pediatric cohort in Delhi, India

**Bireshwar Sinha**[1,2], **Nidhi Goyal**[1], **Mohan Kumar**[1], **Aashish Choudhary**[3], **Alok Arya**[1], **Anitha Revi**[1], **Ankita Dutta**[4], **Deepak More**[4], **Temsunaro Rongsen-Chandola**[1] *

**1** Centre for Health Research and Development, Society for Applied Studies, New Delhi, India, **2** Clinical and public health fellow, DBT/Wellcome India Alliance, Hyderabad, India, **3** Virology Lab, Department of Microbiology, All India Institute of Medical Sciences, New Delhi, India, **4** Clinical and Research Laboratories, Society for Applied Studies, New Delhi, India

* naro@sas.org.in

## Abstract

### Background

Our aim was to estimate the overall and age-specific incidence of lab-confirmed dengue fever using ELISA based assays among children 6 months to 15 years in Delhi.

### Methods

We enrolled a cohort of 984 children aged 6 months to <14 years in South Delhi and followed-up weekly for fever for 24 months or till 15 completed years of child-age. Households of the enrolled children were geo-tagged. NS1, IgM and IgG assays were conducted using ELISA method to confirm dengue fever in children with ≥3 consecutive days of fever. Molecular typing was done in a subset of NS1 positive cases to identify the circulating serotypes.

### Principal findings

We had a total of 1953 person-years (PY) of follow up. Overall, there were 4208 episodes of fever with peaks during June to November. The overall incidence (95%CI) of fever was 215/100 PY (209 to 222). A total of 74/1250 3-day fever episodes were positive for acute dengue fever (NS1 and/or IgM positive). The overall incidence (95%CI) of acute dengue fever was 37.9 (29.8 to 47.6) per 1000 PY; highest among children aged 5 to 10 years (50.4 per 1000 PY, 95% CI 36.5 to 67.8). Spatial autocorrelation analysis suggested a clustering pattern for the dengue fever cases (Moran's Index 0.35, z-score 1.8, p = 0.06). Dengue PCR was positive in 16 of the 24 specimens tested; DEN 3 was the predominant serotype identified in 15/24 specimens.

### Conclusions

We found a high incidence of dengue fever among under 15-year children with clustering of cases in the community. DEN 3 was the most commonly circulating strain encountered. The findings underscore the need for development of affordable pre-vaccination screening strategy as well as newer dengue vaccines for young children while continuing efforts in vector control.

**Data Availability Statement:** All relevant data are within the manuscript and its Supporting Information files

**Funding:** The authors have received no funding for this specific work presented.

**Competing interests:** The authors have declared that no competing interests exist.

## Author summary

South Asian countries especially the Indian subcontinent contributes the highest to the global burden of dengue. The number of dengue cases reported in India is likely an under-estimate of the actual disease burden and there is a clear lack in the availability of population-based data on incidence of dengue in India. In our pediatric cohort of 984 children aged 6 months to 15 years from Delhi, India, we found a high incidence of lab-confirmed dengue fever, with the highest burden among 5 to 10 year old children. Dengue fever was observed to be highest in the post-monsoon months with significant clustering of cases in the community. DEN 3 was the most commonly circulating strain encountered. Given the high burden in children, the findings highlight the need for strengthening efforts to developing newer dengue vaccines for younger children.

## Introduction

The World Health Organization listed dengue as one of the top 10 potential threats to global health in 2019. Global burden of dengue was estimated to be 104 million (95% CI 64 to 159) cases in 2017 [1] with the highest burden in South Asia of 3546.9 cases (95% Uncertainty interval 2128.5 to 5429.5) per 100,000 population [1]. The number of dengue related deaths, globally, is estimated to increase from 6957 (95% UI 7613 to 30091) in 1990 to 40467 (95% UI 17620 to 49778) in 2017 [1]. Clinically apparent dengue can affect all age groups but is most frequently observed in the age group 5–15 years, globally and the highest mortality is in under-five children [2]. Modeling of surveillance data suggest, India contributes to around one-third of the global burden of symptomatic dengue infections [3].

In India, during last two decades, large and frequent dengue outbreaks and urban to rural spread with increasing proportion of severe cases, along with hyper endemicity in urban areas are reported [4]. In 2017, National Vector Borne Disease Control Program (NVBDCP) reported 188,401 laboratory confirmed cases of dengue and 325 deaths through its network of 646 sentinel surveillance hospitals [5, 6]. It is speculated that the number of dengue cases is highly under-reported in India [7]. A recent systematic review in 2018 estimated that the proportion of laboratory confirmed dengue cases on testing of 213,285 clinically suspected patients from 180 studies in India was 38.3% (95% CI 34.8% to 41.8%)[8]. The overall seroprevalence of dengue fever in India based on 7 studies was 56.9% (95% CI 37.5% to 74.4%). The systematic review observed a clear lack in the availability of population-based data on incidence of dengue in India and highlighted the need of community-based studies for burden estimation [8].

Dengue prevention and control has been largely based on principles of vector control. However, currently several new dengue vaccines are in development. In December 2015, the first dengue vaccine (CYD-TDV, or Dengvaxia) was registered and is now licensed in several countries including Mexico and Philippines. The WHO in its 2016 position paper on vaccination against dengue recommends its use only in children 9 years and above in high burden countries. With newer indigenous vaccines being developed, it is imperative to have population-based incidence data on dengue for conducting vaccine studies and designing vaccination strategies. Our study was planned with the primary objective to estimate the incidence of lab-confirmed acute dengue fever using ELISA based assays in children under 15 years of age. In addition, we planned to report the age-categorized incidence in children aged <5 years, 5–10 years, and 10–15 years of age. As an exploratory objective we reported the most common circulating serotype in the population in a subsample.

## Methods

### Ethics statement

The cohort study was approved by the Ethics Review Committee, Centre for Health Research and Development Society for Applied Studies, New Delhi, India in 2017.

### Study population

The study was conducted as a sub-cohort within the Surveillance of Enteric Fever in India (SEFI) cohort among children in Delhi (CTRI/2017/09/009719) [9], in two contiguous blocks of Sangam Vihar, a low-middle income urban and peri-urban neighborhood in the southern district of Delhi [10]. In the study area, more than 80% of the population are Hindu by religion, more than half of the women have secondary school education, with an average household monthly income between 10,000 to 20000 INR, similar to the National average [11].

A door-to-door survey was conducted in the two blocks to line list the households with potentially eligible children. Families were screened to determine likelihood of moving out of the study area and/or adhering to study procedures specially blood specimen collection. Eligible children aged between 6 months and <14 years who were likely to stay in the study area for two years were enrolled consecutively in the cohort from contiguous households until the sample size was attained. All enrolled children were followed up for 24 months or until 15 years of age, whichever was earlier. The upper age limit at enrollment was restricted to <14 years so that each enrolled child in the cohort be followed up for at least 12 months. Prior to screening, written informed consent for participation and storage of specimens for future research was obtained from the parents after explaining study processes. We obtained verbal assent from children aged 7 to 12 years and written assent from children aged above 12 years. (Fig 1).

### Acute febrile illness surveillance

We conducted active weekly contacts either through phone or home visit with at least one face-to-face contact every four weeks to collect information on fever, illness or hospitalizations. A monthly mobile recharge of INR 50 was provided to all enrolled families to encourage early telephonic reporting of fever to the study team. Families were provided with a digital thermometer and fever diary card to document temperature. Fever was defined as caregiver reported or a recorded temperature of ≥38˚C (100.4˚F). Upon reporting of fever, our study team visited the participant's home as early as possible. Participant was visited daily during the fever episode. If fever lasted for ≥3 days, the child was referred to the study fever clinic at Hakeem Abdul Hameed Centenary Hospital (HAHCH) for physician evaluation and blood specimen collection (not later than 6 days of fever onset) to test for enteric fever (blood culture) and dengue fever. Management of the fever episode was as per the advice of the physician in the study fever clinic or any other preferred place of care-seeking. In case the participant visited other health facilities for care-seeking, information related to the fever episode was collected during the weekly contacts.

### Laboratory methods

Blood specimens were transported to Clinical and Research Laboratories, Society for Applied Studies (CRL SAS), New Delhi, within 2 hours of collection. Serum was separated in CRL SAS and used for the laboratory assays for dengue using the automated EVOLIS Twin Plus system (BioRad, California, USA). Dengue NS1 Ag Microlisa, dengue IgM Microlisa and dengue IgG Microlisa (J. Mitra & Co. Pvt. Ltd., New Delhi, India) kits were used.

Primary Cohort, children aged 6 months to <14 years enrolled between Sep 2017 to March 2018 = 6000

↓

Dengue sub cohort, children enrolled between 24 Oct 2017 and 25 Jan 2018 = 984 children

Follow up: 24 months from enrolment or until 15 years of age, whichever was earlier

Total number of Acute febrile illness (AFI) episodes = 4208
Eligible for Dengue testing: AFI episodes with duration >=3 consecutive days = 1504

Blood specimen could not be obtained =254(16.8%)
Refusal by parent = 205
Family not in study area = 46
Current fever episode within ≤14 days of previous episode = 3

Tested for Dengue (NS1, IgM, IgG)
1250 AFI episodes

Positive for Dengue
Total Dengue NS1/IgM positive: Acute Dengue fever = 74
NS1 +ve and/or IgM +ve with IgG -ve: Acute primary dengue fever = 68
NS1 +ve and/or IgM +ve with IgG +ve: Acute secondary dengue fever = 6
Isolated IgG +ve = 29

Dengue serotype identification

Specimens with high OD value for Dengue NS1 >1.8 (n=24) evaluated for molecular typing of DEN viruses
DENV1= 1; DENV2= 0; DENV3= 15; DEN4=0; Undetected = 8

**Fig 1. Study Flowchart.**

To study the circulating serotypes of dengue viruses, the serum specimens positive for NS1 antigen with an optical density (OD) value of $\geq$1.8 were further used for molecular typing. This was based on prior experience that it is difficult to get a positive signal for a specific dengue serotype in multiplex PCR assays in specimens with lower NS1 OD values. The molecular typing of dengue viruses was conducted at the Virology lab of All India Institute of Medical Sciences, New Delhi using the Geno-Sen's dengue 1–4 (Rotor Gene) real time PCR quantitative kit (Genome Diagnostics Pvt. Ltd.).

## Data management

Data was collected on tablets using android application package with built-in Geographic Information Systems (GIS) developed in house by the SEFI team and stored in a secured Amazon cloud-based server. Data was monitored using a dashboard-based system; weekly reports were generated and reviewed. A comprehensive audit trail was maintained. Quality assurance was done by an independent team of experts.

## Statistical analysis

To capture an incidence of acute dengue fever as low as 3 per 100 person-years (PY) of follow up among children under 15 years of age with 95% confidence, 30% relative precision, and 25% loss to follow up a total of 1841 PY of follow up was deemed to be sufficient. Based on our previous experience and considering each child will contribute to around 1.9 years of follow up, a total of 969 eligible children were deemed to be required.

Analysis was done using STATA 16.0 MP (TX, USA). The Stata command "xtset" was used to declare the dataset to be panel data. The characteristics of the study population was presented as proportions (%) and mean (SD). 'Person-year' for each child was calculated from date of enrolment to censorship (end of study, withdrawal, death, or no face-to-face contact for $\geq$90 days). During the follow-up, if a participant was not contactable for $\geq$2 consecutive weeks, the information from the previous 2-weeks from the date of subsequent contact was captured and considered in person-years of follow up and the intervening time period was not considered in analysis.

Fever episodes positive for dengue NS1 and/or IgM were defined as lab-confirmed acute dengue fever episodes. In addition to NS1 and/or IgM, fever episodes that were positive for dengue IgG were identified as acute secondary dengue fever and those negative for IgG were acute primary dengue fever. Children were divided into 3 age categories i.e., 6 months to <5 years, 5 years to <10 years and 10 to 15 years. The age was calculated from date of birth verified from birth certificate or immunization card. The age-specific incidence rates of acute dengue fever or acute febrile illness was calculated as the number of new events in the specific age-interval divided by total number of person-years of follow-up contributed by all children at risk in this interval with allowing for children to move to higher age categories. Confidence intervals were calculated using the "poisson" option in Stata. The cases of lab-confirmed dengue fever were plotted using ArcGIS 10.8 on a base map using Sentinel 2 satellite data with 10 m spatial resolution available from Copernicus Open Access Hub (https://scihub.copernicus.eu/) of 2021 for the study area. After preparing the GIS maps, Moran's Index of spatial autocorrelation was estimated to identify any clustering pattern of the dengue fever cases.

## Results

We followed up a fixed cohort of 984 children from 755 households from the date of first enrollment, 24[th] October 2017 to date of completed follow up, 13[th] February 2020. In this cohort we had a total of 1953 person-years (PY) of follow up, of which, 535, 854 and 564 PY

were contributed by children aged 6 months to <5 years, 5 to <10 and 10 to 15 years, respectively (Fig 1). During this follow-up period a total of 16 episodes of hospitalizations were reported in this pediatric cohort.

## Characteristics of the study population

Around two-third (72%) of the children enrolled belonged to nuclear families with median monthly income of 143 USD; 77% of these households were overcrowded. Piped water from the Delhi municipal corporation (81%) and bottled water (12%) were the major sources of household drinking water (Table 1).

## Acute febrile illness surveillance

In this cohort, there were 4208 episodes of fever with peaks during the months of June to November (Table 2 and Fig 2). The overall incidence (95%CI) of fever was 215/100 PY (209 to

**Table 1. Characteristics of the study population.**

| Baseline characteristics of enrolled children | n/N (%) [a] |
|---|---|
| Sex of the Child: Female | 474/984 (48.17) |
| Age at enrollment | |
| 6 months to < 5years | 342/984 (34.76) |
| 5 years to <10 years | 434/984 (44.11) |
| 10 years to 15 years | 208/984 (21.13) |
| **Sociodemographic parameters** | |
| Type of Family | |
| Nuclear | 540/755 (71.52) |
| Three generation | 47/755 (6.23) |
| Joint | 168/755 (22.25) |
| Family Size: Mean (SD) | 5.83 (2.11) |
| Highest education in family as years of schooling: Mean (SD) | 11.05 (3.17) |
| Type of House [b] | |
| Pucca | 749/755 (99.21) |
| Mixed | 6/755 (0.79) |
| Kuccha | 0/755 (0.00) |
| Overcrowding present (>2.5 persons per living room) | 583/755 (77.22) |
| Separate kitchen available | 535/755 (70.86) |
| Primary fuel used for cooking in household | |
| LPG Gas | 754/755 (99.87) |
| Others | 1/755 (0.13) |
| Monthly income in USD [c] | |
| Median (IQR) | 142.86 (114.29, 214.29) |
| Mean (SD) | 192.76 (159.64) |
| Source of Drinking water | |
| Piped water in household | 609/755 (80.66) |
| Bottled water | 88/755 (11.66) |
| Public tap/standpipe | 34/755 (4.50) |
| Tube well | 7/755 (0.93) |
| Tanker-truck | 16/755 (2.12) |
| Others | 1/755 (0.13) |

[a] This is a cohort of 984 children from 755 households–the denominators are mentioned accordingly

[b] A pucca house is one, which has walls and roof made of bricks, stones packed with cement. Kutcha houses are made of material other than those mentioned above, such as bamboos, mud, grass, reeds, thatch, etc.

[c] 1 US dollar (USD) = 70 Indian Rupee (INR)

[d] Methods of water treatment include boiling, chlorination, and filtration

**Table 2. Incidence of fevers by age-categories in the cohort (n = 984).**

| Age group | Person years of follow up | Number of Fever episodes | Incidence (95%CI) of fever per 100 PY | Number of ≥3-day fever episodes | Incidence (95% CI) of ≥3-day fevers per 100 PY |
|---|---|---|---|---|---|
| 6m to <5y | 535.08 | 3251 | 607.6 (587.5 to 628.2) | 572 | 106.9 (98.4 to 115.9) |
| 5y to <10y | 853.84 | 547 | 64.1 (58.8 to 69.6) | 589 | 68.9 (63.5 to 74.7) |
| 10y to 15y | 563.70 | 410 | 72.7 (65.9 to 80.1) | 343 | 60.8 (54.6 to 67.6) |
| All children | 1952.62 | 4208 | 215.5 (209.1 to 222.0) | 1504 | 77.0 (73.2 to 81.0) |

222), with highest incidence in under-five children 607.6/100 PY (587.5 to 628.2). Median (IQR) duration of a fever episode was 2 (1–4) days. In 34.7% (1460/4208) of the fever episodes an antibiotic was prescribed which was initiated around day 2 (IQR 1 to 3) of fever onset. Oral beta-lactam, cephalosporin and macrolides were the most prescribed antibiotics. Of the 1504 episodes of 3-day fevers, dengue tests were conducted in 1250 episodes (83.1%, Fig 1). The median (IQR) time of blood collection was on day 3 (3 to 4) from onset of fever. In the other 3-day fever episodes (254/1504) dengue tests could not be performed due to refusal by parent for blood collection or family unavailable in the study area.

## Acute dengue fever

A total of 74/1250 3-day fever episodes were positive for acute dengue fever (Dengue NS1 and/ or IgM positive); 68/74 were primary acute dengue fever, and 6/74 were acute secondary dengue fever episodes (Table A in S1 Appendix). The positivity rate (95% CI) for dengue NS1 and/or IgM in our cohort was 5.9% (4.6% to 7.4%). Spatial autocorrelation analysis suggested a clustering pattern for the dengue fever cases in the study area with a 6% likelihood that the observed clustered pattern could be due to random chance (Moran's Index 0.35, z-score 1.8, p = 0.06, Fig 3). We observed higher number of acute dengue fever episodes during August to December (monsoon and post-monsoon seasons) with the highest risk in the month of October. Number of dengue fevers were higher in 2018 compared to 2019 (Fig 2).

Overall incidence (95%CI) of acute dengue fever was 37.9 (29.8 to 47.6) per 1000 PY, primary dengue fever was 34.8 (27.0 to 44.2) per 1000 PY and secondary dengue fever was 3.1

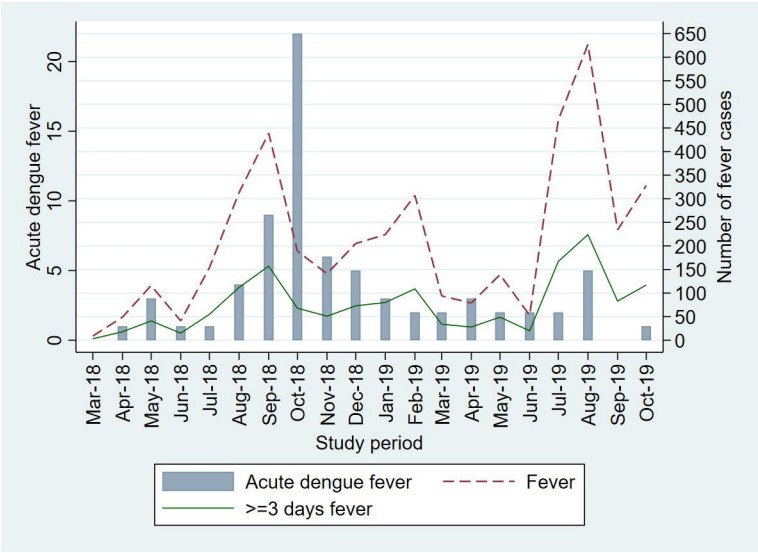

**Fig 2. Acute febrile illness and dengue fever during the study period, by month.**

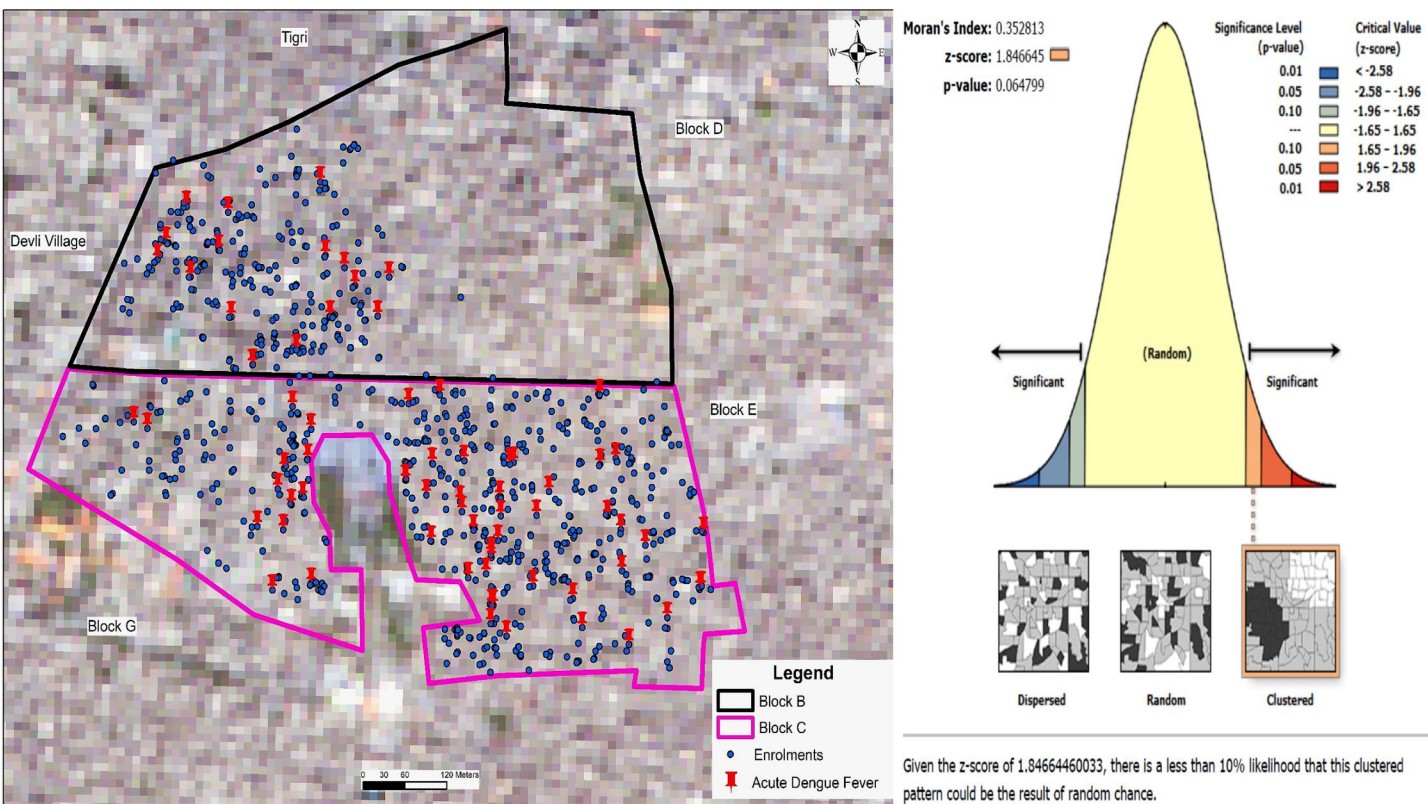

**Fig 3. Geographic distribution of acute dengue fever cases in the study area.** Figure legend: Moran's Index = 0.353, Variance = 0.04, z-score = 1.85, p-value = 0.06. Given the z-score of 1.85, there is a less than 10% likelihood that this clustered pattern could be the result of random chance.

(1.1 to 6.7) per 1000 PY. Incidence of acute dengue fever was observed to be highest among children aged 5 to 10 years (50.4, 95% CI 36.5 to 67.8 per 1000 PY, Table 3). The mean duration of acute dengue fever episode was 6.9 (1.7) days. Only one child had two episodes of acute dengue fever at ages of 5 year 3 months and 6 year 5 months. Two children with acute dengue fever episodes were hospitalized, and mean duration of hospitalization was 8 days. All children with dengue fever recovered; no dengue fever associated deaths were observed (Table B in S1 Appendix).

Of all the acute dengue fever episodes, in 24 cases OD value of dengue NS1 was found to be >1.8 and were subjected to molecular typing of dengue virus (DENV). Dengue PCR was positive in 16 of the 24 specimens (66.7%). The median viral load of dengue in the specimens measured by PCR was estimated to be 138 (IQR 9.25 to 505) x $10^4$ copies/ml. In the tested specimens, the predominant serotype identified was DEN3 in 62.5% (15/24), followed by DEN1 in 4.2% (1/24). We did not identify any DEN2 or DEN4 serotype (Fig 1).

**Table 3. Incidence of dengue fever by age-categories in the cohort (n = 984).**

| Age group in years | Person years | Acute dengue fever | | Acute dengue fever–primary | | Acute dengue fever–secondary | |
|---|---|---|---|---|---|---|---|
| | | No. of episode | Incidence (95%CI) per 1000 PY | No. of episode | Incidence (95%CI) per 1000 PY | No. of episode | Incidence (95%CI) per 1000 PY |
| 6m to <5y | 535.08 | 20 | 37.38 (22.83 to 57.72) | 19 | 35.51 (21.38 to 55.45) | 1 | 1.87 (0.05 to 10.41) |
| 5y to <10y | 853.84 | 43 | 50.36 (36.45 to 67.83) | 40 | 46.85 (33.47 to 63.79) | 3 | 3.51 (0.73 to 10.27) |
| 10y to 15y | 563.70 | 11 | 19.51 (9.74 to 34.91) | 9 | 15.97 (7.30 to 30.31) | 2 | 3.55 (0.43 to 12.82) |
| All children | 1952.62 | 74 | 37.90 (29.76 to 47.57) | 68 | 34.83 (27.04 to 44.15) | 6 | 3.07 (1.13 to 6.69) |

Beyond dengue, in the 1250 AFI episodes, 18 were found to be positive for Salmonella Typhi, 7 were found to be positive for Salmonella Paratyphi and 7 were found to be positive for Chikungunya.

## Discussion

Our pediatric cohort showed high burden of acute dengue fever in low-middle income urban settings in Delhi. The incidence of dengue fever was highest among children aged 5 to 10 years. All children recovered without any associated major complications or death. Acute dengue fever episodes were higher in the months of August to December; more dengue fever episodes were recorded in 2018 against 2019. We observed a clustering pattern in the dengue fever cases in the community. In our study sample, DEN3 was the most encountered strain.

The estimates of dengue fever in children from our study seems to be similar or higher compared with community-based cohorts from other countries [12–15]. In a prospective cohort of 3000 Latin American children aged 9–16 years followed up during 2010–2011 the reported incidence of dengue fever was 41 (95% CI 32 to 54) per 1000 person-years [13]. The incidence of dengue fever in the Nicaraguan pediatric cohort study of 5,545 children aged 2–14 years was 16.1 (95% CI 14.5 to 17.8) per 1,000 person-years [15]. In 2012–13, a prospective cohort of subjects ≥6 months old in Cebu City, Philippines reported incidence of symptomatic dengue to be 16.2 per 1000 person-years [14]. Between November 2008 and January 2010, in a cohort of 800 children from urban Colombo, Sri Lanka the incidence of dengue fever per 1000 children per year was 33.8 (95% CI 22.4 to 48.8) [16].

But there are limited community-based studies that reported incidence of dengue fever in Indian children. Similar to our effort, the Vellore site of the SEFI cohorts reported the annual incidence rate of dengue in 2017–2018 to be 49.5 per 1000 PY in children aged 6 months to 14 years with fever >3 days [17]. Using the serial sero-survey methodology during 2014–2016, a study in a rural area of Pune, Maharashtra estimated the incidence rate of primary dengue infections as 54.2/1000 PY (95% CI 43.0 to 67.3) among children aged 5 to 15 years [18]. Despite some heterogeneity due differences in methodology, pooled estimates of our study along with above two studies indicate that the population-based incidence of symptomatic dengue fever may vary from around 35 to 60 per 1000 PY among Indian children under 15 years of age across different geographies [12–15]. However, multicentric studies with improved diagnostics and standard methodologies across sites may be helpful to generate more precise country-specific estimates.

For diagnosis of dengue fever, rt-PCR is currently considered as the gold standard although NS1 has high sensitivity and specificity. But even rt-PCR may have its own challenges. It has been seen that rt-PCR may be false negative in cases where dengue virus RNA is detected using transcription-mediated amplification (TMA) technique, and where NS1 is found to be positive. Therefore, it has been suggested that NS1 antigen assay is a valuable method for diagnosis of dengue independent of rt-PCR [19]. In limited-resource settings, some researchers even consider NS1 ELISA assay to be superior to rt-PCR due to its high sensitivity, low cost, ease of performance and rapidity [19, 20]. NS1 in combination with IgM assay may offer the most sensitive and cost-effective diagnostic modality for dengue diagnosis [20]. Previous longitudinal studies in Latin America [13] and in India [17] have used NS1 along with IgM assay to estimate incidence of dengue fever, which is similar to our study. Nonetheless, we acknowledge that rt-PCR in addition to NS1 and IgM could enhance the accuracy of the estimates.

The National Centre for Disease Control reported a higher number of cases in 2018 compared to 2019 in Delhi as was also observed in our study [6]. The higher burden of dengue cases as observed during the post-monsoon months of September to December can be

attributed to climatic conditions such as mean temperatures and precipitation levels which shortens the extrinsic incubation period of the dengue virus [17, 21]. Beyond the seasonal peak, we observed cases of dengue throughout the year suggesting sustained endemic transmission similar to other Southeast Asian countries [22]. All four strains of the dengue virus are known to circulate during epidemics in India [23]. The community-based study from Vellore, Tamil Nadu, in 2017–18 reported DEN 1 as most common subtype in their population [17]. DEN 1 and 2 has been previously reported to be the commonest circulating strains in North India [24]. We found DEN 3 as the most common strain in our study sample. Our study was not powered to determine the serotype-specific incidence of dengue and therefore should be interpreted with caution. Nonetheless, the findings corroborate to a recent study in Delhi [25] and may indicate an epidemiological shift. The changing trend of the virus type underscores the importance of continuous disease surveillance.

The study findings have important implications. The high burden of dengue fever specially in the 5 to 10 year old children highlights their vulnerability given the current WHO recommendation to use the available dengue vaccine only in those with a documented past infection or in children above 9 years of age in select areas of high seroprevalence >80% [26]. In addition to continued efforts towards development of newer dengue vaccines and evaluating its effectiveness in younger children, it is crucial to develop affordable pre-vaccination screening strategy with high sensitivity and specificity. The significant clustered pattern of cases in our study area may be related to local mosquito breeding patterns in these sites and highlight the importance of promoting vector surveillance and control.

This study, to the best of our knowledge is the first report related to community-based burden of dengue fever among children under 15 years of age estimated using a pediatric cohort design in low-middle income settings in Delhi, North India. Our study has several limitations. First, our strategy for blood culture precludes capture of dengue fevers with <3-day febrile illness. Second, there were challenges to conduct blood tests in 20% (254/1250) of the eligible 3-day fever episodes. Based on the 5.9% dengue positivity rate in our cohort, it is possible that around 14/254 of these missed 3-day fever episodes could have tested positive for dengue, giving an overall projected incidence rate of 45/1000 PY. Third, the study was not powered for ascertainment of severity of disease. Fourth, given the median duration of a fever episode of 2 days, a biweekly contact would have allowed better capture of fevers at the cost of operational feasibility. We used a combination of active weekly contacts along with incentivization in form of monthly phone recharge for promoting passive reporting of fever. With our approach, we may have missed some fever episodes as many of these households have a common mobile phone, not always available with the primary caregiver of the child to report fever. Fifth, although rt-PCR is the gold standard for dengue diagnosis, our case ascertainment was based on NS1 and/or IgM positivity. But, given the high sensitivity and specificity (>95%) of the kits compared to rt-PCR, we believe this is not a major source of bias. Lastly, our study population is restricted to children <15 years belonging to low-middle income urban area in Delhi, North India and may not be generalizable to adults or other populations with different characteristics or dissimilar geographies.

In conclusion, this cohort study provides robust evidence on the high burden of dengue fever among children in urban, peri urban low middle-income populations in Delhi, India and can be applicable to other similar settings with contextual relevance. Given that India has large variations in geographical and climatic conditions, multicentric studies with standard methodology across sites may help to capture the within country variation in dengue incidence. We think improved methods for disease surveillance along with research for development of newer vaccines particularly for children, together can help to bring down morbidity and mortality related to dengue in our country.

## Supporting information

**S1 Appendix.** Table A: Distribution of different dengue assay results and its classification. Table B: Clinical characteristics and outcomes of acute primary, acute secondary and past dengue infection.
(DOCX)

**S1 Strobe checklist. STROBE Statement—Checklist of items that should be included in reports of cohort studies.**
(DOCX)

**S1 Data. Dataset used for analysis.**
(XLSX)

## Acknowledgments

We thank the whole team at Clinical and research laboratories at Society for Applied Studies team and the field team for the Tier 1 SEFI study at CHRD SAS for their intensive efforts. We are grateful for the guidance and support provided by Dr. Nita Bhandari, Senior Scientist and Director at Centre for Health Research and Development Society for Applied Studies, New Delhi.

## Author Contributions

**Conceptualization:** Bireshwar Sinha, Nidhi Goyal, Temsunaro Rongsen-Chandola.

**Data curation:** Bireshwar Sinha, Mohan Kumar.

**Formal analysis:** Bireshwar Sinha, Mohan Kumar.

**Investigation:** Bireshwar Sinha, Nidhi Goyal, Aashish Choudhary, Deepak More, Temsunaro Rongsen-Chandola.

**Methodology:** Bireshwar Sinha, Aashish Choudhary, Anitha Revi, Ankita Dutta, Deepak More.

**Project administration:** Nidhi Goyal, Alok Arya, Anitha Revi, Ankita Dutta.

**Resources:** Nidhi Goyal, Temsunaro Rongsen-Chandola.

**Supervision:** Nidhi Goyal, Alok Arya, Deepak More, Temsunaro Rongsen-Chandola.

**Validation:** Bireshwar Sinha, Ankita Dutta, Deepak More.

**Visualization:** Anitha Revi.

**Writing – original draft:** Bireshwar Sinha.

**Writing – review & editing:** Bireshwar Sinha, Nidhi Goyal, Mohan Kumar, Aashish Choudhary, Deepak More, Temsunaro Rongsen-Chandola.

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
