## [Decision Letter · Decision Letter 0]

22 Dec 2021

Dear Dr Rongsen-Chandola,

Thank you very much for submitting your manuscript "Incidence of lab-confirmed Dengue fever in a pediatric cohort in North India" for consideration at PLOS Neglected Tropical Diseases. As with all papers reviewed by the journal, your manuscript was reviewed by members of the editorial board and by several independent reviewers. In light of the reviews (below this email), we would like to invite the resubmission of a significantly-revised version that takes into account the reviewers' comments. In particular I would note that one of the reviewers raised very substantive criticisms, and the manuscript will only be suitable for publication if these can be adequately addressed. 

We cannot make any decision about publication until we have seen the revised manuscript and your response to the reviewers' comments. Your revised manuscript is also likely to be sent to reviewers for further evaluation.

Sincerely,

Gregory Gromowski

Associate Editor

Francis Jiggins

Deputy Editor

Reviewer's Responses to Questions

**Key Review Criteria Required for Acceptance?**

**Methods**

-Are the objectives of the study clearly articulated with a clear testable hypothesis stated?

-Is the study design appropriate to address the stated objectives?

-Is the population clearly described and appropriate for the hypothesis being tested?

-Is the sample size sufficient to ensure adequate power to address the hypothesis being tested?

-Were correct statistical analysis used to support conclusions?

-Are there concerns about ethical or regulatory requirements being met?

Reviewer #1: 1) The authors need to describe more on the analysis, especially the stats analysis.

2) What are the diagnosis of secondary and primary infection, why are they probable?

3) I am curious about the severity of the disease. Why do the author not classify the severity?

Reviewer #2: The diagnosis of dengue in this study depends on commercial NS1 testing and IgM testing, which are inadequate within the first 3 days of fever to capture all cases. Ideally all samples should have undergone RT-PCR in addition to NS1 testing. Therefore the data presented here may still be an underestimate. Selecting a subset of NS1 positive cases for RT-PCR (again this sample is very small) is rather non-sensical as NS1 negative patients can be positive with RT-PCR. 

It seems authors have done the RT-PCR not as a diagnostic test for dengue, but to determine the infecting serotype. Again if this is the case a sample size calculation is needed to confidently say that DENV-3 predominates the current epidemic and this small sample is representative of all dengue cases in "North India" to come to this conclusion.

On top of the above limitations, the study only samples a limited geographical region in South Delhi predominantly from low socioeconomic background. Therefore this data is not representative of "North India" or even greater Delhi.

Reviewer #3: The authors have studied the incidence of Dengue fever in cohort of children in low income urban community in Delhi. This was a sub-cohort of the study on surveillance of enteric fever in India done at the same site. The sample size was adequate for the estimating the incidence which was the main objective .The age group studied does not include children above 15 years. 

 Most of the infections were primary infections . The sample recruited was not adequate to study the incidence of severe forms of dengue fever.

**Results**

-Does the analysis presented match the analysis plan?

-Are the results clearly and completely presented?

-Are the figures (Tables, Images) of sufficient quality for clarity?

Reviewer #1: 1) There is no information of missing data

2) How many did participants withdraw from the study or miss the follow ups due to moving out the area?

Reviewer #2: The above limitations highlighted in methods makes the estimates in results unreliable

Reviewer #3: It is not clear if other causes of fever were excluded. The clinical details of any accompanying symptoms is not mentioned. This is important as a significant proportion were prescribed antibiotics. 

 Cases which had only IgM positivity with negative NS1 antigen could have had dengue fever in recent past as IgM can stay in circulations for a couple of months. It will be better to give the number of cases which were in this category .

**Conclusions**

-Are the conclusions supported by the data presented?

-Are the limitations of analysis clearly described?

-Do the authors discuss how these data can be helpful to advance our understanding of the topic under study?

-Is public health relevance addressed?

Reviewer #1: (No Response)

Reviewer #2: The limitations are not properly acknowledged. With this data the authors cannot fufil the aims of their study since the overall dengue incidence is likely to be unerestimated without RT-PCR for all samples.

Reviewer #3: Yes .Limitations have been discussed. Conclusion stated in the last paragraph are not a part of this study since vector surveillance has not been done.

**Editorial and Data Presentation Modifications?**

Reviewer #1: Minor comments

1) Please consistently use small letter for dengue such as "several new Dengue vaccines",

Reviewer #2: (No Response)

Reviewer #3: minor revision

**Summary and General Comments**

Reviewer #1: (No Response)

Reviewer #2: Overall, this article has limited value to an international audience as it only focusses on low socioeconomic groups in a limited geographical region in South Delhi. To say this is representative of dengue incidence in "Northern India" (as mentioned in the title), is misleading . This data may not even be representative of greater metropolitan Delhi (since not all socipoeconomic groups are reprersented) and on top of that may underestimate the true incidence of dengue due to issues in diagnosis highlighted above.

Reviewer #3: It is a well written article with public health importance. It emphasizes the burden of dengue fever in low income community of North India. The authors report change in circulating strain to DEN 3 which highlights the need for continued surveillance.

PLOS authors have the option to publish the peer review history of their article (what does this mean?). If published, this will include your full peer review and any attached files.

Reviewer #1: No

Reviewer #2: No

Reviewer #3: No
---

## [Decision Letter · Decision Letter 1]

8 Mar 2022

Dear Dr Rongsen-Chandola,

Thank you very much for submitting your manuscript "Incidence of lab-confirmed Dengue fever in a pediatric cohort in North India" for consideration at PLOS Neglected Tropical Diseases. As with all papers reviewed by the journal, your manuscript was reviewed by members of the editorial board and by several independent reviewers. The reviewers appreciated the attention to an important topic. Based on the reviews, we are likely to accept this manuscript for publication, providing that you modify the manuscript according to the review recommendations. 

When making your revisions, please be sure to address points from Reviewer 2 regarding "North India" in the Title/text and removing information from Table 1 that is less relevant for dengue. Also, be sure to address the comment from Reviewer 3 about age mismatches in parts of the manuscript.

Sincerely,

Gregory Gromowski

Associate Editor

Francis Jiggins

Deputy Editor

When making your revisions, please be sure to address points from Reviewer 2 regarding "North India" in the Title/text and removing information from Table 1 that is less relevant for dengue. Also, be sure to address the comment from Reviewer 3 about age mismatches in parts of the manuscript.

Reviewer's Responses to Questions

**Key Review Criteria Required for Acceptance?**

**Methods**

-Are the objectives of the study clearly articulated with a clear testable hypothesis stated?

-Is the study design appropriate to address the stated objectives?

-Is the population clearly described and appropriate for the hypothesis being tested?

-Is the sample size sufficient to ensure adequate power to address the hypothesis being tested?

-Were correct statistical analysis used to support conclusions?

-Are there concerns about ethical or regulatory requirements being met?

Reviewer #1: (No Response)

Reviewer #2: I have reviewed this paper before (Reviewer 2). I do not think the reviewers have adequately addressed my concerns. I accept their rebuttal on diagnostic tests. However many other concerns still remain which are listed below

1. The study title still misleadingly refers to "North India". This samples is not representative of North India. While the authors state that "The population characteristics of our study area in South Delhi in terms of socioeconomic status, literacy and religion is representative of around 70-80% of similar populations in India", there is no reference or evidence to back this claim.

2. The serotyping data lacks adequate power to cme to any conclusion and I disagree with their notion of not needing a sample size calculation

3. The cohort was designed for surveillance of enteric fever and some of the data seems to be collected for that purpose with little relevance for dengue. For example in Table 1 the variables mentioned has little relevance to dengue

4. The data only refers to a specific geographical area of Delhi and has little relevance for an international audience.

Reviewer #3: The main objective of the study was to ascertain the incidence of lab confirmed dengue. The invetigators followedup a cohort of children for episodes of acute febrile illness . The population is clearly described and meticulously followed up

The methods are described in detail. .

**Results**

-Does the analysis presented match the analysis plan?

-Are the results clearly and completely presented?

-Are the figures (Tables, Images) of sufficient quality for clarity?

Reviewer #1: There is a mismatch of children age in different parts of the manuscript such as 5-15, 5-< 14...

There are a lot of self plagiarism as shown in the attached file.

Reviewer #2: Please see comments above

Reviewer #3: The results are presented clearly and figures and images are satisfactory.

**Conclusions**

-Are the conclusions supported by the data presented?

-Are the limitations of analysis clearly described?

-Do the authors discuss how these data can be helpful to advance our understanding of the topic under study?

-Is public health relevance addressed?

Reviewer #1: (No Response)

Reviewer #2: Please see comments above

Reviewer #3: The conclusion are supported by the data. The limitations are described . The public health implications are discussed.

**Editorial and Data Presentation Modifications?**

Reviewer #1: (No Response)

Reviewer #2: (No Response)

Reviewer #3: The authors have revised the document as per modifications suggested.

**Summary and General Comments**

Reviewer #1: (No Response)

Reviewer #2: (No Response)

Reviewer #3: Well designed study .

PLOS authors have the option to publish the peer review history of their article (what does this mean?). If published, this will include your full peer review and any attached files.

Reviewer #1: No

Reviewer #2: No

Reviewer #3: No

Figure Files:

Data Requirements:

Reproducibility:

References

---

## [Editor Report · Decision Letter 2]

15 Mar 2022

Dear Dr Rongsen-Chandola,

We are pleased to inform you that your manuscript 'Incidence of lab-confirmed dengue fever in a pediatric cohort in Delhi, India' has been provisionally accepted for publication in PLOS Neglected Tropical Diseases.

Best regards,

Gregory Gromowski

Associate Editor

Francis Jiggins

Deputy Editor

---

## [Editor Report · Acceptance letter]

4 Apr 2022

Dear Dr Rongsen-Chandola,

We are delighted to inform you that your manuscript, "Incidence of lab-confirmed dengue fever in a pediatric cohort in Delhi, India," has been formally accepted for publication in PLOS Neglected Tropical Diseases.

Best regards,

Shaden Kamhawi

co-Editor-in-Chief

Paul Brindley

co-Editor-in-Chief
